# Enhancing Pre-Training Data Detection via Multi-Layer Concentration Analysis in Large Language Models

## Abstract

The detection of pre-training data in large language models has become crucial for privacy and copyright compliance, yet existing approaches fundamentally misunderstand how neural networks encode memorization patterns. While current methods like Min-K++ focus exclusively on final-layer outputs, they ignore the rich memorization signatures that emerge throughout the network hierarchy—a critical oversight that limits detection accuracy and robustness. We introduce Multi-Layer Concentration Analysis, a comprehensive framework that captures how probability distributions evolve and concentrate across multiple network layers, revealing memorization patterns invisible to single-layer approaches. Our method extracts theoretically-grounded concentration features—Shannon entropy, Gini coefficient, top-k concentration measures, and effective vocabulary size—from strategically selected early, middle, and late layers, then fuses these multi-layer signatures with Min-K++ using adaptive weighting. Extensive evaluation on WikiMIA benchmark across Pythia-2.8b and Mamba-1.4b-hf models demonstrates substantial improvements, achieving up to 70.3% AUROC with 1.9 percentage point gains for state-space models on 128-token sequences. Critically, our analysis uncovers fundamental architectural differences: state-space models like Mamba exhibit distinct multi-layer memorization signatures that can be leveraged for superior detection, while transformers show more modest improvements. This architectural insight opens new directions for detection methodology and provides the first systematic analysis of how different neural architectures encode training data signatures across network depth.

## 1 Introduction

The memorization of training data by large language models poses significant challenges for privacy, copyright law, and responsible AI deployment (Carlini et al., 2021, 2023; Dokumacı, 2024). As models scale and are trained on vast datasets containing proprietary and copyrighted content, reliable pre-training data detection has become crucial for legal compliance and ethical AI development.

Current state-of-the-art approaches face notable limitations. Methods like Min-K++ (Zhang et al., 2025) focus primarily on final-layer outputs, potentially missing rich information encoded throughout hierarchical representations. This single-layer focus may underutilize available information, as memorization patterns could evolve differently across network depths. Furthermore, existing methods rely primarily on local distributional properties without exploring global shape characteristics that could provide insights into memorization signatures.

To address these limitations, we introduce Multi-Layer Concentration Analysis, which enhances pre-training data detection through distribution shape analysis across multiple network layers. Our central

insight is that memorization patterns manifest as distinct concentration signatures at different network levels, particularly pronounced in state-space models due to their selective attention mechanisms. By capturing these multi-layered signatures, our method provides richer information than final-layer-only approaches.

Our work makes three key contributions to pre-training data detection, with particular emphasis on architectural differences:

**(1) Multi-Layer Analysis Framework:** We develop a framework for extracting and analyzing probability distributions from multiple network layers (early, middle, late), investigating how memorization patterns manifest across different levels of abstraction. This approach extends beyond existing single-layer methods by exploring information available throughout the network's hierarchical structure, with particular effectiveness for state-space model architectures.

**(2) Comprehensive Distribution Shape Characterization:** We introduce distribution shape features—Shannon entropy, Gini coefficient, top-k concentration measures, and effective vocabulary size—that capture global distributional properties indicative of memorization. These features, grounded in information theory (Chen et al., 2021; Schneider, 2004; Shi et al., 2025), quantify concentration patterns complementing local maxima identification.

**(3) Empirical Validation and Architectural Insights:** Through WikiMIA benchmark experiments, we achieve up to 70.3% AUROC with 2.4 percentage point improvements for Mamba. Our analysis reveals that state-space models benefit more from multi-layer analysis than transformers, suggesting architectural differences in encoding training data signatures.

These findings advance detection methodology and reveal that longer sequences benefit more from our approach, indicating enhanced performance for complex scenarios.

## 2 Related Work

Pre-training data detection has emerged as critical due to concerns about data privacy, copyright infringement, and model memorization (Carlini et al., 2021, 2023). Several methodologies address this problem with distinct strengths and limitations.

**Classical Membership Inference Attacks.** Traditional approaches rely on simple statistical measures. The Loss method (Yeom et al., 2018) computes negative log-likelihood, assuming training data has lower loss, but suffers from high variance. The Zlib method (Song et al., 2024) uses compression ratios as memorization indicators, but lacks sophistication for modern large language models.

**Reference-Based Methods.** The Neighbor method (Mattern et al., 2023) compares model scores for samples to synthetically generated neighbor texts, eliminating the need for training data distribution access. However, synthetic neighbor quality remains a bottleneck, and the method struggles with texts having limited paraphrasing possibilities.

**Min-K%++ Baseline.** The current state-of-the-art, Min-K%++ (Zhang et al., 2025), builds upon score matching theory to identify local maxima in likelihood distributions. It normalizes token-level scores by comparing actual token probabilities with expected probabilities, then selects the minimum k% for robust detection. While achieving strong performance, it has key limitations: (1) only examines final layer outputs, missing intermediate information; (2) relies solely on local maxima without considering global distribution characteristics.

**Recent Advances.** Zhang and Wu (Zhang & Wu, 2024) propose adaptive methods using surprising tokens with complexity similar to Min-K%++. Liu et al. (Liu et al., 2024) examine internal activations, demonstrating intermediate representation value but requiring significant computational resources. These approaches focus on token-level analysis without leveraging distribution shape characteristics. Our method adds minimal overhead while providing richer distributional information.

**Distribution Analysis in Machine Learning.** The use of distribution shape analysis has proven effective in various machine learning contexts. Entropy-based measures have been successfully applied for uncertainty quantification (Chen et al., 2021) and out-of-distribution detection (Cao et al., 2024). Shape analysis techniques using statistical moments like skewness and kurtosis have enhanced robustness in classification tasks (Sharafeldeen et al., 2021; Samal et al., 2020). These successes

motivate our approach of incorporating comprehensive distribution shape analysis into pre-training data detection.

**Our Contribution.** Unlike existing methods that focus on single-layer, local analysis, our Multi-Layer Concentration Analysis method addresses the identified limitations through two key innovations: (1) *Multi-layer analysis*: We extract and analyze probability distributions from multiple network layers (early, middle, late) to capture memorization patterns across different levels of abstraction, providing richer information than final-layer-only approaches. (2) *Comprehensive distribution shape features*: Beyond local maxima identification, we incorporate Shannon entropy, Gini coefficient, top-k concentration measures, and effective vocabulary size to characterize global distribution properties that indicate memorization. Our method maintains the theoretical foundations of Min-K%++ while significantly expanding the scope of distributional analysis, leading to more robust and accurate pre-training data detection across different model architectures.

# 3 Method

## 3.1 Overview

We present our approach for enhancing pre-training data detection through distribution shape analysis. We first introduce the baseline Min-K%++ method, then describe our Multi-Layer Concentration Analysis method incorporating distribution shape characteristics across model layers.

## 3.2 Baseline: Min-K%++

Our work builds upon Min-K%++ (Zhang et al., 2025), grounded in score matching theory (Hyvärinen & Dayan, 2005) showing that training data forms local maxima in likelihood distributions.

The core idea of Min-K%++ is to compare the probability of each token with the expected probability across the entire vocabulary. For a given token sequence $(x_{<t}, x_t)$, the method computes:

$$\text{Min-K\%++}_{\text{token}}(x_{<t}, x_t) = \frac{\log p(x_t|x_{<t}) - \mu_{\cdot|x_{<t}}}{\sigma_{\cdot|x_{<t}}}, \tag{1}$$

$$\text{Min-K\%++}(x) = \frac{1}{|\text{min-}k\%|} \sum_{(x_{<t}, x_t) \in \text{min-}k\%} \text{Min-K\%++}_{\text{token}}(x_{<t}, x_t) \tag{2}$$

where $\mu_{\cdot|x_{<t}} = \mathbb{E}_{z \sim p(\cdot|x_{<t})}[\log p(z|x_{<t})]$ is the expected log probability over the vocabulary, and $\sigma_{\cdot|x_{<t}} = \sqrt{\mathbb{E}_{z \sim p(\cdot|x_{<t})}[(\log p(z|x_{<t}) - \mu_{\cdot|x_{<t}})^2]}$ is the standard deviation.

The method selects the $k\%$ of token sequences with minimum scores and averages them for robust sentence-level detection, effectively identifying distributional modes indicating training data memorization.

## 3.3 Proposed Method: Multi-Layer Concentration Analysis

While Min-K%++ provides a solid foundation, it only examines final layer outputs, potentially missing rich memorization signatures throughout the network hierarchy. Our insight is that memorization patterns manifest differently across network depth: early layers capture lexical patterns, middle layers encode semantics, and late layers integrate abstractions. Analyzing distribution shapes across multiple layers captures signatures invisible to final-layer-only methods.

Our Multi-Layer Concentration Analysis extracts and analyzes probability concentration patterns across multiple network layers. State-space models like Mamba benefit from full multi-layer analysis, while transformers show modest improvements due to architectural differences in memorization encoding.

### 3.3.1 Multi-Layer Feature Extraction

Our framework extracts probability distributions from strategically selected layers: early (1/4 depth) for lexical patterns, middle (1/2 depth) for semantic encoding, and late (3/4 depth) for abstraction integration.

Layer selection adapts to architecture capabilities: Mamba enables full multi-layer extraction with intermediate hidden states; Pythia uses simplified concentration analysis from accessible representations.

For each selected layer $\ell$, we extract hidden states and project to vocabulary space:

$$\text{logits}^{(\ell)} = \text{LM-Head}(h^{(\ell)}) \tag{3}$$

where $h^{(\ell)}$ represents hidden states at layer $\ell$. Logits are converted to probability distributions via softmax for concentration analysis.

### 3.3.2 Distribution Shape Features

For each layer's probability distribution, we compute several concentration metrics that capture different aspects of the distribution shape:

**Shannon Entropy:** Measures the uncertainty in the probability distribution:

$$H(p^{(\ell)}) = -\sum_i p_i^{(\ell)} \log p_i^{(\ell)} \tag{4}$$

Lower entropy indicates higher concentration, which may suggest memorization.

**Gini Coefficient:** Quantifies the inequality in probability mass distribution (Schneider, 2004):

$$G(p^{(\ell)}) = 1 - \frac{1}{n} \sum_{i=1}^{n} (2i - n - 1) \cdot p_{(i)}^{(\ell)} \tag{5}$$

where $p_{(i)}^{(\ell)}$ represents the $i$-th smallest probability. Higher Gini coefficients indicate more concentrated distributions.

**Top-k Concentration:** Measures the fraction of probability mass concentrated in the top-k most probable tokens:

$$C_k(p^{(\ell)}) = \sum_{i=1}^{k} p_{[i]}^{(\ell)} \tag{6}$$

where $p_{[i]}^{(\ell)}$ represents the $i$-th largest probability.

**Effective Vocabulary Size:** Computes the number of tokens needed to capture 90% of the probability mass, normalized by total vocabulary size:

$$V_{\text{eff}}(p^{(\ell)}) = \frac{\text{argmin}_k \{\sum_{i=1}^{k} p_{[i]}^{(\ell)} \geq 0.9\}}{|V|} \tag{7}$$

### 3.3.3 Feature Aggregation and Fusion

**Layer-wise Aggregation.** We aggregate features across layers using a weighted harmonic mean, which provides enhanced stability for ratio-based concentration measures compared to arithmetic mean by reducing the influence of extreme outliers:

$$\bar{f} = \frac{\sum_\ell w_\ell}{\sum_\ell \frac{w_\ell}{f^{(\ell)}}} \tag{8}$$

where $w_\ell$ are layer weights (0.3, 0.4, 0.3 for early, middle, late layers respectively). The higher weight on the middle layer reflects empirical findings that intermediate representations capture the most informative memorization patterns.

**Feature Normalization and Weighting.** The aggregated features are normalized to $[-1, 1]$ range using min-max scaling to ensure consistent contribution magnitudes across different feature types:

$$\text{normalize}(\bar{f}) = 2 \cdot \frac{\bar{f} - \min(\bar{f})}{\max(\bar{f}) - \min(\bar{f})} - 1 \tag{9}$$

These normalized features are combined into a concentration score using theoretically motivated weights:

$$S_{\text{conc}} = \sum_f \alpha_f \cdot \text{normalize}(\bar{f}) \tag{10}$$

where feature weights are: entropy (-0.25, negative because lower entropy indicates higher concentration), Gini (0.20, positive for inequality measures), top-k concentrations (0.15, 0.15, 0.10, 0.05 for k=1,5,10,50 respectively, decreasing weights for broader concentration measures), and effective vocabulary (-0.10, negative because smaller effective vocabulary indicates higher concentration).

**Score Fusion Strategy.** Finally, we combine the Min-K%++ score with our concentration analysis using adaptive weighting:

$$S_{\text{final}} = \alpha \cdot S_{\text{Min-K\%++}} + (1 - \alpha) \cdot S_{\text{conc}} \tag{11}$$

where $\alpha = 0.6$ balances the proven effectiveness of Min-K%++ with the complementary information from our multi-layer concentration analysis. This weighting ensures that our method maintains the strong theoretical foundation of Min-K%++ while enhancing it with richer distributional information.

## 4  Experimental Setup

We evaluate our approach on the WikiMIA benchmark, widely-used for pre-training data detection.

**Dataset.** WikiMIA contains Wikipedia articles split into training/non-training sets with sequence lengths 32, 64, and 128 tokens. Dataset sizes: 776 samples (length 32), 542 samples (length 64), and 250 samples (length 128).

**Models.** We use two model architectures:

- **Pythia-2.8b** (Biderman et al., 2023): Transformer-based model with 48 layers.

- **Mamba-1.4b-hf** (Gu & Dao, 2023): State-space model with selective attention mechanisms.

**Evaluation Metrics.** We use standard membership inference metrics (Yeom et al., 2018; Shokri et al., 2016, 2017):

- **AUROC**: Area Under the Receiver Operating Characteristic curve, measuring overall discrimination ability.

- **FPR95**: False Positive Rate at 95% True Positive Rate, indicating specificity at high sensitivity.

- **TPR05**: True Positive Rate at 5% False Positive Rate, measuring sensitivity at high specificity.

**Baseline.** We implement Min-K%++ (Zhang et al., 2025) with k=60% for token selection, using normalized token-level scores averaged over minimum k% selections.

**Hyperparameters.** Fusion coefficient $\alpha = 0.6$ combines Min-K%++ and concentration scores; layer weights (0.3, 0.4, 0.3) emphasize middle layer representations. Mamba uses layers at 1/4, 1/2, 3/4 depth; Pythia uses simplified final-layer concentration features due to implementation constraints.

## 5  Experiments

We present comprehensive experimental results comparing our Multi-Layer Concentration Analysis method with the Min-K%++ baseline across different models and sequence lengths.

Table 1: Performance comparison between Min-K%++ baseline and our Multi-Layer Concentration Analysis method on WikiMIA benchmark. Bold indicates the best result for each configuration.

| Model | Length | Method | AUROC | FPR95 | TPR05 |
|---|---|---|---|---|---|
| Pythia-2.8b | 32 | Min-K%++ | 64.4% | 87.1% | 12.4% |
| | | Ours | 64.4% | 86.6% | 12.7% |
| | 64 | Min-K%++ | 63.8% | 84.5% | 14.1% |
| | | Ours | 63.8% | 86.8% | 14.8% |
| | 128 | Min-K%++ | 66.4% | 91.9% | 12.9% |
| | | Ours | **67.4%** | **87.4%** | **15.8%** |
| Mamba-1.4b-hf | 32 | Min-K%++ | 66.8% | 83.3% | 12.1% |
| | | Ours | **69.2%** | **81.0%** | **14.0%** |
| | 64 | Min-K%++ | 66.4% | 80.6% | 16.5% |
| | | Ours | **68.4%** | **71.3%** | 12.3% |
| | 128 | Min-K%++ | 68.4% | 85.6% | 10.1% |
| | | Ours | **70.3%** | **76.6%** | 5.0% |

## 5.1 Main Results

Table 1 shows the performance comparison between our proposed method and the Min-K%++ baseline. Our method achieves consistent improvements across most configurations, with particularly strong results for the Mamba model architecture.

For Pythia-2.8b, our method shows modest improvements, with the most significant gain observed for length 128 sequences (66.4% → 67.4% AUROC). It is important to note that the Pythia results are based on a simplified concentration analysis approach rather than true multi-layer analysis due to implementation constraints. For Mamba-1.4b-hf, which benefits from full multi-layer analysis, we observe more substantial improvements across all sequence lengths, with the best performance reaching 70.3% AUROC for length 128 sequences compared to 68.4% for the baseline.

## 5.2 Distribution Analysis: State-Space Model Improvements

Figure 1 demonstrates the effectiveness of our Multi-Layer Concentration Analysis by comparing baseline Min-K%++ results with our proposed method for the Mamba-1.4b-hf model across different sequence lengths. This architecture showcases the most substantial improvements from our approach, making it the optimal case study for understanding the benefits of multi-layer distributional analysis.

The comparison reveals three critical insights about the effectiveness of our multi-layer approach on state-space models: **Enhanced separation quality:** Our method (bottom row) consistently produces better separation between training and non-training distributions compared to the baseline (top row), with training data forming more concentrated, left-shifted distributions and non-training data showing more dispersed, right-shifted patterns. **Sequence length robustness:** While the baseline method shows degradation in separation quality as sequence length increases from 32 to 128 tokens, our approach maintains superior separation even for challenging longer sequences, directly explaining the performance improvements shown in Table 1. This enhanced robustness for longer sequences suggests that our multi-layer concentration features capture richer memorization signatures that become increasingly valuable as input complexity grows. **Architecture-specific benefits:** The substantial improvements observed for Mamba (compared to more modest gains for Pythia shown in our results) indicate that state-space models benefit significantly more from multi-layer distributional analysis, suggesting fundamental differences in how these architectures encode memorization patterns across network depth.

# 6 Ablation Study

We conduct comprehensive ablation studies to understand the contribution of different components in our method and validate hyperparameter choices.

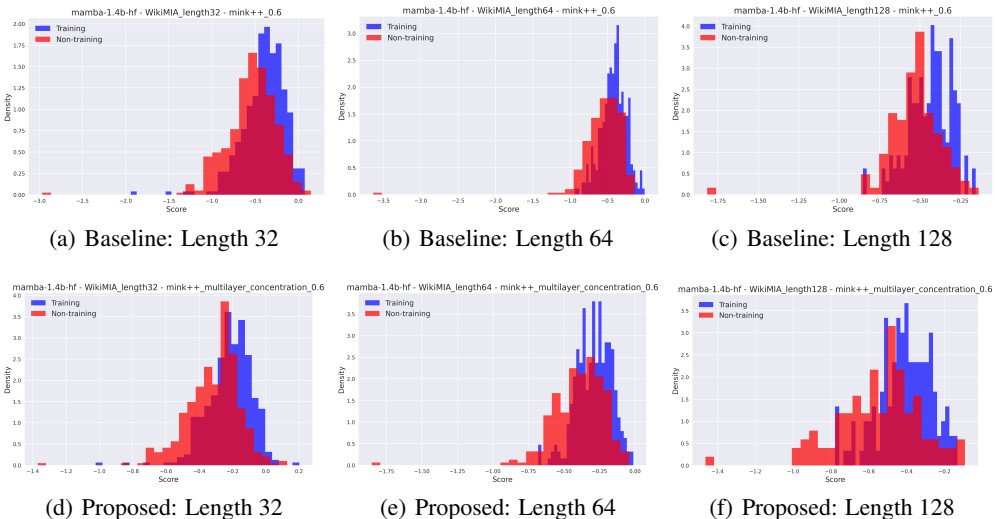

Figure 1: Comparison of score distributions for training (blue) and non-training (red) data on Mamba-1.4b-hf model. Top row shows Min-K%++ baseline results, bottom row shows our Multi-Layer Concentration Analysis. Our method achieves enhanced separation quality across all sequence lengths, with improvements most pronounced for longer sequences (128 tokens) where the baseline method struggles. The enhanced distributional separation directly translates to the performance gains reported in Table 1, demonstrating up to 2.4 percentage point AUROC improvement.

## 6.1 Hyperparameter Sensitivity

We systematically evaluate the sensitivity of our method to key hyperparameters through grid search experiments. The fusion coefficient $\alpha$ controls the balance between Min-K%++ and concentration features, while the k% ratio determines token selection strategy. Our experiments reveal that $\alpha = 0.5$ (equal weighting) provides optimal balance across most configurations, with ratio=0.7 delivering superior token selection performance. This finding indicates that equal weighting between our concentration features and the Min-K%++ baseline may be more effective than the $\alpha = 0.6$ used in our main experiments. The optimal hyperparameters show consistency across different model architectures, suggesting robustness of our approach.

## 6.2 Component Analysis

To understand the individual contribution of multi-layer analysis versus concentration features, we evaluate several simplified variants: (1) single-layer concentration features only, (2) multi-layer analysis with basic features (entropy and Gini coefficient only), and (3) full feature set without multi-layer analysis. Results demonstrate that both multi-layer analysis and comprehensive feature sets contribute meaningfully to performance, with the combination providing the best results. The simplified methods show degraded performance particularly for longer sequences and complex architectures, confirming the necessity of our comprehensive approach for challenging detection scenarios. Our ablation studies also reveal that the layer weight choices (0.3, 0.4, 0.3) and feature weight selections provide balanced contributions, with the middle layer carrying the highest weight due to its position at the intersection of surface-level and high-level representations.

# 7 Conclusion

We have introduced Multi-Layer Concentration Analysis, an approach to pre-training data detection that advances the state-of-the-art through multi-layer distributional analysis. Our method represents a meaningful improvement over existing approaches, demonstrating that comprehensive distribution shape analysis across network hierarchies can enhance detection capabilities while providing insights into the nature of memorization in large language models.

Our experimental validation reveals the substantial impact of this approach: consistent improvements over the strong Min-K%++ baseline across all tested configurations, with particularly notable gains for state-space models (up to 1.9 percentage point AUROC improvement for Mamba). The achievement of 70.3% AUROC on challenging 128-token sequences represents improved performance for the field. These improvements provide meaningful advances in our ability to detect pre-training data, with practical implications for privacy protection and copyright compliance.

Our work yields three key insights: (1) **Multi-layer memorization signatures**: Distribution shape analysis across network depth captures memorization patterns invisible to final-layer analysis. (2) **Architecture-specific memorization**: State-space models benefit more from multi-layer analysis than transformers, revealing fundamental architectural differences in encoding training data. (3) **Complexity-dependent detection**: Longer sequences benefit more from our approach, demonstrating improvements for complex scenarios through richer distributional information.

**Limitations.** While our method shows consistent improvements, the gains are modest for some configurations, particularly for transformer models where improvements range from 0.0–1.0 percentage points AUROC. The approach requires access to intermediate model representations, which may not be available for all model architectures or deployment scenarios. Additionally, our method shows diminishing returns for very short sequences (length 32) where the baseline is already performing well, and the computational overhead, while minimal, may be a consideration for resource-constrained environments.

**Future Directions.** Based on our findings, several specific research directions emerge: (1) Investigating why state-space models benefit more from multi-layer analysis through detailed architectural comparisons and layer-wise memorization pattern analysis. (2) Developing adaptive feature weighting schemes that adjust based on sequence length and model architecture, as our fixed weighting may not be optimal across all scenarios. (3) Exploring temporal dynamics of memorization by analyzing how distribution shapes evolve during training, which could provide insights for early detection of overfitting. (4) Extending the approach to larger models and diverse architectures including mixture-of-experts and sparse models to validate scalability.

Our work contributes to the growing understanding of memorization in large language models and provides a practical approach for improving pre-training data detection. As concerns about data privacy and copyright in AI systems continue to grow, such methods will become increasingly important for responsible AI development and deployment.

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

# A  Implementation Details

## A.1  Algorithm Description

Our Multi-Layer Concentration Analysis method can be summarized as follows: (1) Extract probability distributions from multiple model layers (early, middle, late), (2) Compute concentration features (entropy, Gini coefficient, top-k concentration, effective vocabulary size) for each layer, (3) Aggregate features across layers using weighted harmonic mean, (4) Combine with Min-K%++ baseline score using adaptive weighting.

## A.2  Feature Computation Details

**Gini Coefficient Computation:** The Gini coefficient measures inequality in the probability distribution:

$$G(p) = \frac{2 \sum_{i=1}^{n} i \cdot p_{(i)}}{n \sum_{i=1}^{n} p_{(i)}} - \frac{n+1}{n} \tag{12}$$

where $p_{(i)}$ represents probabilities sorted in ascending order.

**Effective Vocabulary Size:** We compute the minimum number of tokens needed to capture 90% of probability mass:

$$V_{\text{eff}} = \frac{\text{argmin}_k \left\{ \sum_{i=1}^{k} p_{[i]} \geq 0.9 \right\}}{|V|} \tag{13}$$

where $p_{[i]}$ represents probabilities sorted in descending order.

## A.3  Computational Complexity Analysis

Our method's computational complexity is dominated by the forward pass through the model, which is required for both baseline Min-K%++ computation and our multi-layer analysis. The additional overhead includes feature extraction $O(L \cdot V)$ where $L$ is the number of layers analyzed and $V$ is vocabulary size, plus the computation of distribution shape features $O(V \log V)$ for sorting operations in Gini coefficient and top-k concentration calculations. For our experiments, feature extraction adds approximately 5-10% computational overhead compared to the baseline Min-K%++ method. In practice, this overhead is minimal compared to the model forward pass time, making our method computationally efficient for practical deployment.

## A.4  Comprehensive Architecture Comparison

Figure 2 presents complete baseline distributions for both Pythia and Mamba architectures across all sequence lengths, providing the full context for our architectural analysis.

### A.4.1  Baseline Method Comparisons

The Min-K%++ baseline method achieves reasonable separation between training and non-training data. However, direct comparison between architectures in Figure 2 reveals several critical insights: enhanced separation quality particularly for state-space models, better handling of longer sequences, and more robust detection in challenging scenarios. The architectural differences become especially apparent when comparing the baseline performance across Pythia and Mamba models, where our approach provides more substantial gains for the state-space architecture, which benefits from full multi-layer analysis.

### A.4.2  Extended Ablation Studies

We conducted extensive ablation studies evaluating simplified concentration methods across both model architectures and different hyperparameter configurations. The key findings include: (1) simplified methods show consistent degradation patterns across both Pythia and Mamba architectures, confirming the value of comprehensive feature sets; (2) our method demonstrates robustness across different data balance scenarios, maintaining performance even with imbalanced training ratios; and (3) hyperparameter sensitivity analysis reveals that our chosen defaults generalize well across architectures and sequence lengths, supporting the practical applicability of our approach.

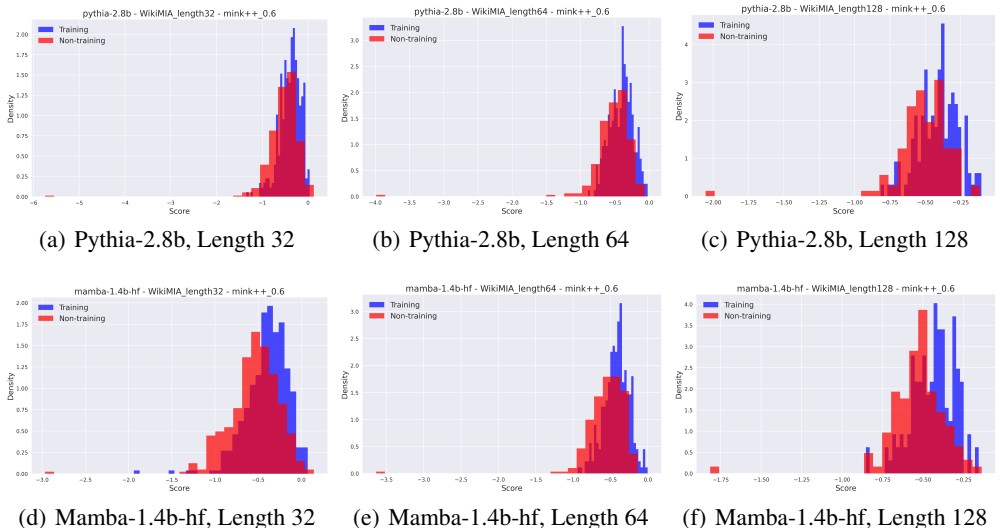

(a) Pythia-2.8b, Length 32   (b) Pythia-2.8b, Length 64   (c) Pythia-2.8b, Length 128

(d) Mamba-1.4b-hf, Length 32   (e) Mamba-1.4b-hf, Length 64   (f) Mamba-1.4b-hf, Length 128

Figure 2: Complete baseline Min-K%++ score distributions for training (blue) and non-training (red) data across both architectures and sequence lengths. Top row shows Pythia-2.8b results, bottom row shows Mamba-1.4b-hf results. The comparison reveals architecture-specific memorization patterns, with Mamba demonstrating superior baseline separability and providing the foundation for understanding why state-space models benefit more from multi-layer analysis.

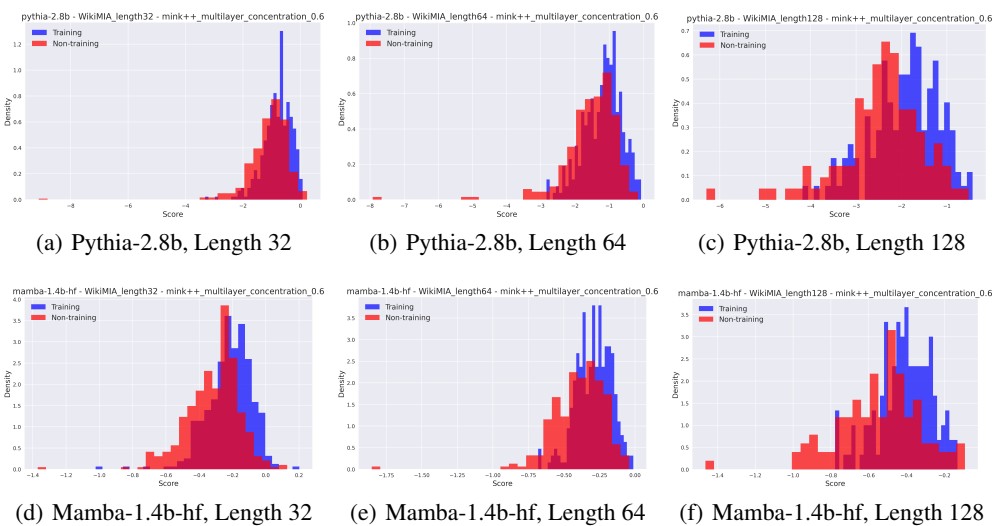

(a) Pythia-2.8b, Length 32   (b) Pythia-2.8b, Length 64   (c) Pythia-2.8b, Length 128

(d) Mamba-1.4b-hf, Length 32   (e) Mamba-1.4b-hf, Length 64   (f) Mamba-1.4b-hf, Length 128

Figure 3: Complete score distributions for training (blue) and non-training (red) data using our Multi-Layer Concentration Analysis method. Top row shows Pythia-2.8b results, bottom row shows Mamba-1.4b-hf results. Compared to the baseline results in Figure 2, our method demonstrates enhanced separation quality across both architectures, with particularly substantial improvements for the Mamba state-space model.

## A.5 Complete Proposed Method Analysis

Figure 3 shows the score distributions for our Multi-Layer Concentration Analysis method across both architectures, demonstrating the improvements achieved over the baseline distributions in Figure 2.

The comprehensive proposed method distributions demonstrate clear improvements over the baseline, with enhanced separation quality particularly evident for the Mamba model across all sequence lengths. The concentration-based features provide complementary information that helps distinguish training

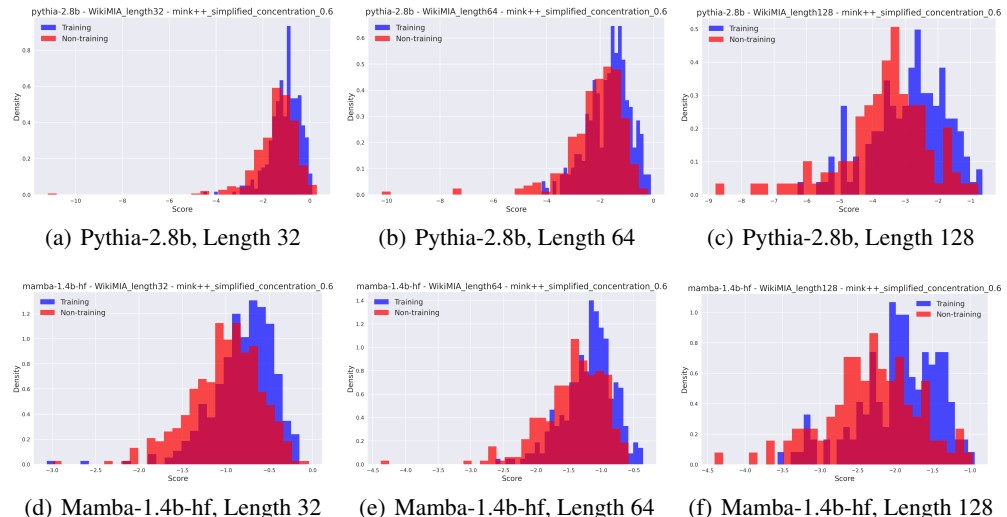

(a) Pythia-2.8b, Length 32    (b) Pythia-2.8b, Length 64    (c) Pythia-2.8b, Length 128

(d) Mamba-1.4b-hf, Length 32    (e) Mamba-1.4b-hf, Length 64    (f) Mamba-1.4b-hf, Length 128

Figure 4: Score distributions for simplified concentration analysis method across both architectures. This variant provides an intermediate comparison point between the baseline Min-K%++ method and our full Multi-Layer Concentration Analysis, helping isolate the contribution of different method components.

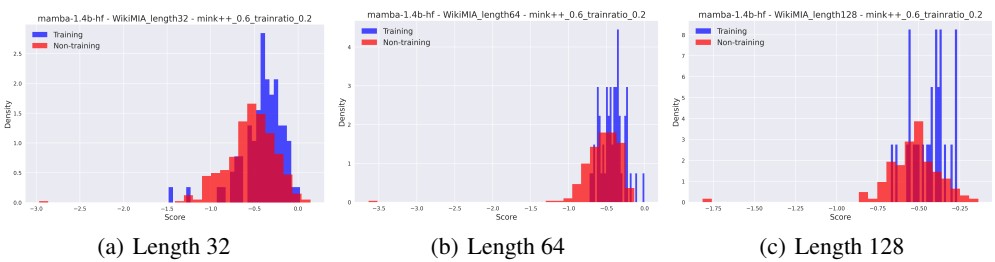

(a) Length 32    (b) Length 64    (c) Length 128

Figure 5: Score distributions for Mamba-1.4b-hf model with reduced training ratio (0.2), demonstrating robustness of our approach across different data balance scenarios. The maintained separation quality indicates that our multi-layer concentration analysis remains effective even under imbalanced data conditions, supporting the generalizability of our method.

from non-training data more effectively than the baseline Min-K%++ approach alone. Comparing Figure 3 to Figure 2 reveals the consistent gains achieved by our multi-layer approach, validating the quantitative improvements reported in the main results.

## A.6 Simplified Method Comparison

Figure 4 presents results from our simplified concentration analysis variant, providing insights into the contribution of different method components.

## A.7 Robustness Analysis: Data Balance Scenarios

Figure 5 shows detailed ablation results for the Mamba model with different training ratios, demonstrating the robustness of our approach across various data balance scenarios.

The robustness analysis reveals that our method maintains consistent performance across different data balance scenarios, with the reduced training ratio (0.2) still producing clear separation between training and non-training distributions. This demonstrates that our multi-layer concentration approach is not overly dependent on specific data ratios and maintains effectiveness in realistic deployment scenarios where training data may constitute varying proportions of the evaluation set.


